# MIND: A Multi-Source Data Fusion Scheme for Intrusion Detection in Networks

**DOI:** 10.3390/s21144941

**Published:** 2021-07-20

**Authors:** Naveed Anjum, Zohaib Latif, Choonhwa Lee, Ijaz Ali Shoukat, Umer Iqbal

**Affiliations:** 1Department of Computing, Riphah International University, Faisalabad 38000, Pakistan; naveed45us@gmail.com (N.A.); i.shoukat@riphahfsd.edu.pk (I.A.S.); umeriqbal@riphahfsd.edu.pk (U.I.); 2Department of Computer Science, Hanyang University, Seoul 04763, Korea; lee@hanyang.ac.kr

**Keywords:** data fusion, network intrusion detection systems, anomaly detection, machine learning, ensemble learning

## Abstract

In recent years, there is an exponential explosion of data generation, collection, and processing in computer networks. With this expansion of data, network attacks have also become a congenital problem in complex networks. The resource utilization, complexity, and false alarm rates are major challenges in current Network Intrusion Detection Systems (NIDS). The data fusion technique is an emerging technology that merges data from multiple sources to form more certain, precise, informative, and accurate data. Moreover, most of the earlier intrusion detection models suffer from overfitting problems and lack optimal detection of intrusions. In this paper, we propose a multi-source data fusion scheme for intrusion detection in networks (*MIND*) , where data fusion is performed by the horizontal emergence of two datasets. For this purpose, the Hadoop MapReduce tool such as, Hive is used. In addition, a machine learning ensemble classifier is used for the fused dataset with fewer parameters. Finally, the proposed model is evaluated with a 10-fold-cross validation technique. The experiments show that the average *accuracy*, *detection rate*, *false positive rate*, *true positive rate*, and *F-measure* are *99.80%*, *99.80%*, *0.29%*, *99.85%*, and *99.82%* respectively. Moreover, the results indicate that the proposed model is significantly effective in intrusion detection compared to other state-of-the-art methods.

## 1. Introduction

There are billions of users who are connected through thousands of autonomous systems over the Internet globally. The world has witnessed the exponential growth of Internet traffic for many years. This massive upsurge in the network traffic contains data from many sources, which are diverse in nature. Notably, this data may bring different types of anomalies, which can affect network security [1]. It can be observed in Figure 1, which is plotted based on the data of Skybox Security [2] and Redscam Report [3], that the Internet users and network attacks are growing at the same pace. According to the graph, more than 18,000 new attacks were detected in 2020, which is 5.13% more than the previous year.

A wide range of technologies such as user authentication, data encryption techniques, and firewalls are implemented to avoid these issues. However, the analysis of such technologies is not enough. In order to overcome the limitation of these mechanisms, different network intrusion detection systems (NIDS) are used to analyze the network packets more deeply compared to traditional mechanisms for intrusion detection [1] and intrusion tolerant [4] systems. However, these systems are designed for homogeneous scenarios and are unable to detect anomalies from heterogeneous sources. Furthermore, due to massive dimensions and huge volumes of data, there are many challenges, which include: (i) These systems are complex and require more resources in terms of storage and computing; (ii) these systems face challenges due to irrelevant and redundant data; (iii) it is challenging to detect zero-day attacks; and (iv) the low detection and high false alert rate is another challenge [5].

To overcome these issues in NIDS, the data fusion technique is a promising solution [6]. Data fusion is an evolving technique that is implemented in network intrusion detection to resolve the heterogeneity of network data [7]. Data is combined from multiple sources to make decisions about situations, activities, and events in this technique. It provides more informative, consistent, and accurate information than the original raw data that is uninformative, inconsistent, and uncertain. The data fusion scheme for future technologies (e.g., Internet of Things (IoT)) can be very interesting because these devices are more vulnerable, heterogeneous, and have more security threats. The data fusion can be performed at the decision level, feature level, and data layer. The data fusion is applied to military (situation assessment and surveillance) and commercial (manufacturing, remote sensors, and robotics) applications.

To this end, in this paper, we used feature level fusion which is used to merge the data coming from multiple sources and provides a more comprehensive view of data analysis. Since relational database management system (RDBMS) fails to handle significant data storage issues, Hadoop comes to rescue these hitches that is an open-source distributed storage system and runs on commodity machines. Datasets are merged with the help of the Hive tool, which executes by using the Hadoop MapReduce framework [8]. The fused dataset is then evaluated by using machine learning algorithms. The ensemble technique of machine learning is preferred as it uses multiple weak learners to build strong learners [9,10,11]. The bagging technique of ensemble learning is used, which is least affected by the variance as compared to other ensemble learning techniques [12].

The major contributions of this work are the following:To perform data fusion among multi-sources datasets (NSL-KDD and UNSW-NB15) for more certain and informative data;To apply fused data to machine learning algorithm (i.e., K-Nearest Neighbor (KNN) as a base classifier for bagging technique) to check performance after fusion.

The rest of the paper is organized as follows: Background is discussed in Section 2 and Section 3 reviews the related work. Section 4 presents the proposed approach. Experimental setup is described in Section 5. Evaluation and results are discussed in Section 6. Finally, Section 7 concludes this work.

## 2. Background

In this section, we discuss a brief background of NIDS and data fusion techniques before delving into the details of this paper.

There are two primary objectives of an IDS which are the following:(i) to monitor the network systems and (ii) to generate an alarm in case of any abnormal activity. The first objective of monitoring is considered an input to the IDS, whereas the second objective is output. The monitoring can be performed with the help of statistical information from network traffic, packet header, and content information at the network level, whereas system class traces, process behavior, and application logs are extracted from the system level. The output against these parameters is generated in the form of binary or multilevel indication. In binary indication, the traffic is classified as normal or abnormal traffic. The severity and different types of attacks are obtained in multilevel indications. These alarms are helpful to take actions against intrusions and mitigate them; however, a false alarm is another challenge. Moreover, due to significant network traffic, the NIDS introduces complexity and require more resources in terms of computation and storage.

Data fusion is a promising technology where data from multiple sources are combined to make better decisions to overcome these issues. This process deals with the data correlation, association, and combination from different sources to obtain optimal estimation, timely assessment, and the significance of network threats. It can be understood that, with the help of the human cognitive process, actions to do with the evaluation of the situation, decision making, and taking actions based on the fused information of different sensory organs can commence. The data fusion can be applied at three different layers: decision layer, feature layer, and data layer. The decision layer fusion combines the decision of various processing units to provide the final decision. The feature-level fusion helps to reduce the features of preprocessed data. Feature selection plays a vital role by removing irrelevant features in the high dimensional dataset and enhances the model performance. The feature selection procedure is broadly separated into filter, wrapper, and embedded categories [13,14]. Finally, the data layer fusion is used to integrate the raw data from different sources for better understanding. It is also known as low-level fusion.

## 3. Related Work

Several studies propose different mechanisms for intrusion detection in the computer networks. A light-weight algorithm is proposed in [15] for a real-time intrusion detection model based on a deep belief network (DBN) and support vector machine (SVM). The high dimensional data reduction is achieved by using DBN. The sliding window method uses pattern behavior to mine the abnormal data. This pattern behavior library is formed to maintain the abnormal data by using regular model updating. After using the DBN method for feature reduction for high dimensional data, the SVM-based model is trained for IDS. The experiment is performed on the CICIDS 2017 dataset. This approach lacks the data fusion technique and uses only one dataset for model evaluation resulting in an overfitting problem. Furthermore, the accuracy (ACC) and false-positive rate (FPR) are not discussed.

In [16], authors introduce a new semi-self-taught (SST) network intrusion detection system that is based on semi-supervised discriminant autoencoder (SSDA), which requires a minor human intervention. SSDA was implemented through denoising auto-encoder with a c-mean algorithm for class identification. The experiment is based on the CSE-CIC-IDS2018 dataset. This may cause an overfitting problem due to a single dataset. Data fusion is also lacking in this process.

Authors propose the novel XGBoost-DNN model for network intrusion detection in [17]. Firstly, data preprocessing is performed for data normalization. Secondly, the XGBoost method is used for feature selection in high-dimensional data. Finally, binary classification is performed by using a deep neural network (DNN). The learning frequency is increased using the Adam optimizer (AO) during model training. Experiments are performed using the NSL-KDD dataset. This method also suffers from the overfitting problem due to the single dataset. Model evaluation and data fusion are also lacking in this process.

The novel semantic re-encoder and deep learning model (SRDLM) for IDS is discussed in [18]. In this study, the authors propose a model for network anomaly detection by using massive semantic coding space along with word order negligence. The semantic re-encoding has performance limitations with network traffic, however, this limitation is improved by combining it with the deep learning method. The deep learning framework ResNet, based on convolutional neural networks (CNN), is used in this method because of its generalization ability for intrusive unknown network traffic. The experiment is performed on benchmark NSL-KDD and Hduxss_data1.0 datasets. This study shows good performance against the Hduxss_data1.0 dataset with improved results against the NSL-KDD dataset. This study lacks data fusion and feature selection techniques as each dataset is used separately for model evaluation. The FPR evaluation parameter is not calculated against both datasets.

In [19], authors introduce a novel approach for intrusion detection by combining hybrid sampling and deep hierarchical networks. First, one-side selection (OSS) and synthetic minority over-sampling technique (SMOTE) are used to develop a composed dataset for training the model. This process reduces the training time of the model and data preprocessing is performed for complex networks. Second, it builds a model classifier based on a hierarchical network by combining CNN and bi-directional long short-term memory (BiLSTM). UNSW-NB15 and NSL-KDD datasets have been used for model evaluation. The data fusion technique is not used in this article and each dataset is used separately for model evaluation. Moreover, FPR is not considered in this study.

The authors propose a novel intrusion detection framework (DT-EnSVM) in [20] by merging ensemble learning with the data transformation method. First, ratio transformation is applied to get a new and well-balanced dataset. Second, SVM-based classifiers are selected for model training. Finally, the non-mixture method is used to combine these classifiers by building an ensemble learning-based model. NSL-KDD and Kyoto 2006+ datasets are used for experimentation. This study lacks the data fusion technique as each dataset is used separately for the experiment. Moreover, the true positive rate (TPR) and F-Measure parameters are missing.

The adaptive ensemble learning-based model is discussed in [21]. In this study, the authors highlight the key benefits of the ensemble approach by taking different base classifiers. Five machine learning classifiers (i.e., Decision Tree, Support Vector Machine, Logistic Regression, K-Nearest Neighbor, and Random Forest) are used in the ensemble technique. NSL-KDD dataset is used for the experimentation. This article has an overfitting problem. Feature selection and data fusion are also missing in this study. This study does not consider the false-positive rate.

In [22], authors address the training issues of feed-forward neural network (FNN) using locust swarm optimization (LSO), which is an optimization algorithm of the meta-heuristic field. The authors claim that the joint venture of FNN and LSO (FNN-LSO) improves the overall performance of IDS. This work aims to design an IDS based on the evolutionary algorithm (EA) using FNN and LSO. NSL-KDD and UNSW-NB15 datasets are used for experimentation. Feature selection and data fusion are lacking in this article. The authors propose a new model for NIDS in [23] based on a hierarchical neural network that integrates the convolutional neural network (CNN) as LeNet-5 architecture and long short-term memory (LSTM) neural network. This work uses CICIDS 2017 and UTC datasets for experiments. Feature selection and data fusion are missing in this article. Furthermore, the false-positive rate is not considered.

The authors in [24] introduce a new hybrid artificial bee colony (ABC) and artificial fish swarm (AFS) techniques for intrusion detection to differentiate the normal and abnormal network activities. First, the dataset is separated into a small subset using the FCM (C-mean clustering) method. Second, feature selection is performed by using the correlation-based feature selection (CFS) method. Finally, the classification and regression tree (CART) method is used to employ if-then rules to separate regular traffic from anomalies. NSL-KDD and UNSW-NB15 datasets are used to evaluate the classification model. Data fusion techniques are lacking in this article. Moreover, the TPR and F-Measure evaluation parameters are not considered.

In [25], authors propose a fusion scheme to aggregate and merge the real time data from different sensors to detect the man-in-the-middle (MiTM) attacks in power systems. The authors exploit supervised, unsupervised, and semi-supervised learning with a focus on cyber physical systems to detect the false alarm rate. It also provides a comparison between various algorithms. However, this study does not consider all the evaluation metrics.

The authors in [26] exploit decision level data fusion scheme for the denial of service (DoS) attacks. The decisions of basic probability (BP) neural network and D-S evidence theory are combined for the improved results. Moreover, the conflicts between different decision is also focused. Some of the evaluation metrics are discussed, however, most of the metrics are ignored.

Conclusively, many studies are available in the literature for detecting network intrusions, as shown in Table 1. Some of these works use single dataset for this purpose which causes the over-fitting problem. The data fusion scheme is missing in most of the studies where authors use multiple datasets. Moreover, few works do not perform the feature selection, which can degrade the overall performance because of the high dimensionality of data. Finally, some basic evaluation parameters (i.e., FPR, TPR, and ACC) are missing in the results of some studies.

## 4. Proposed Method

In this section, we discuss our proposed approach *MIND* where we have four major steps which are dataset selection, feature selection, data fusion, and machine learning implementation as shown in Figure 2. Firstly, two public datasets NSL-KDD [27] and UNSW-NB15 [28] are selected for model building. Secondly, the relevant data features of NSL-KDD and UNSW-NB15 datasets are selected based on literature study [29]. Thirdly, the data fusion (feature level) process is carried out and horizontally merges the datasets to produce a single uniform and comprehensive dataset using the Hadoop MapReduce tool (i.e., Hive). Finally, data preprocessing is performed on a fused dataset to convert characters into numeric values as machine learning-based models only accept numeric values for their input. In addition, the bagging technique of ensemble learning (also known as Bootstrap Aggregation) is used to build the model for finding intrusions in a fused dataset. Moreover, the proposed model is evaluated through a confusion matrix using 10-fold cross-validation. The step-by-step working of the proposed method is mention below.

### 4.1. Dataset Selection

Dataset plays a significant role in the intrusion detection system for model assessment. Real-time network traffic data cannot be acquired for research purposes due to privacy problems, therefore, researchers are intensely reliant on public datasets [30]. Several public datasets are available for network IDS such as KDD CUP99 [31], NSL-KDD [27], Kyoto 2006 [32], UNSW-NB15 [28], NGIDS-DS [33], ISOT [34], TRAIbID [35], and CICIDS 2017 [36]. For this study, NSL-KDD and UNSW-NB15 are selected for the experiments. In order to perform the data fusion, at least two datasets are required. It is also worth mentioning here that one primary requirement to perform fusion is to have at least one similar column in two different datasets. Since only these two datasets have similar columns in the literature, we therefore selected these datasets for this work.

NSL-KDD is a recommended dataset that is intensively used in literature. In addition, it is a relatively old dataset and does not contain new attacks. It is an enhanced form of the KDD CUP 99 dataset where duplicate instances are removed. This dataset contains 148,517 total instances in which 125,973 are used for training and 22,544 are used for testing purposes. It comprises 41 attributes, including numeric, binary, and nominal values. This dataset includes two types of classes for records as attack and normal. The intrusion instances are categorized as remote to local (R2L), denial of service (DoS), user 2 root (U2R), and Prob.

On the other hand, UNSW-NB15 is a relatively new dataset and contains modern-day attacks. UNSW-NB15 is generated at the Australian Center for Cybersecurity (ACCS) by using the IXIA PerfectStorm tool. This dataset consists of 257,673 total instances with 82,332 test instances and 175,341 for training purposes. It comprises 49 attributes, including nominal, integer, float, and binary values. Similar to NSL-KDD, this dataset is also comprised of both attack and regular records. The attack is categorized as Worm, Shellcode, Reconnaissance, Generic, Exploits, DoS, Backdoor, Analysis, and Fuzzers.

### 4.2. Feature Selection

The machine learning-based model exhibits better performance by removing redundant data and irrelevant features from the dataset. NIDS does not need all the attributes of high-dimensional NSL-KDD and UNSW-NB15 datasets. The feature selection is broadly separated into the filter, wrapper, and embedded categories [14]. The filter method assigns a weight value to the features in the training dataset. The wrapper method is a lengthy process where machine learning algorithms are used for each possible combination. The embedded approach is a combination of filter and wrapper methods. Another technique used for feature selection is the ensemble approach, which produces efficient results by combining various machine learning algorithms and improving classification accuracy.

The feature selection in *MIND* is motivated by [29] where a twofold method is proposed and a novel framework based on data analytics lifecycle is presented to build an intrusion detection system. In addition, ensemble techniques are used to identify the relevant features for intrusion detection. First, the framework consists of data discovery, data preparation, model planning, model building, and model evaluation components to implement the data analytics lifecycle for detecting intrusions. Second, the ensemble method is carefully implemented, which combines multiple models instead of one model to achieve an optimum output. This ensemble approach consists of two steps which are (1) the construction of ensemble components by joining the subset combination policy taking correlation, information, consistency, and distance as feature evaluation procedures for feature selection. (2) Combining the output of ensemble components using subset and ranking combination methods. This ensemble method is evaluated using classification accuracy and diversity between the member of the ensemble and the stability of the ensemble to data variation. Ten features are selected out of 41 from NSL-KDD and eight are selected out of 49 features extracted from UNSW-NB15 datasets. Table 2 and Table 3 shows the acknowledged features of NSL-KDD and UNSW-NB15 datasets, correspondingly.

### 4.3. Data Fusion

The data fusion procedure is achieved by using correlation, integration, estimation, and a combination of data imminent from single or multiple sources. In order to achieve data fusion in this paper, we used relational algebra, which is helpful to provide a joining between different datasets. Let: X = $\left\{x_{1},x_{2},x_{3},\ldots ,x_{n}\right\}$ are the features of dataset${}_{1}$ and Y = $\left\{y_{1},y_{2},y_{3},\ldots ,y_{n}\right\}$ are the features of dataset${}_{2}$. There are different relational operations which are the following: select, project, and rename. In this study, we used the select operation where the tuples are selected on the basis of given conditions for selection and can be represented as follows: $\sigma_{p}\left(r\right)$ where σ is the predicate, *r* is the relation name of the table, and *p* shows the preposition logic. In project operation, all attributes of input relations are eliminated except those which are mentioned in the projection list. Finally, the attributes of a relation are renamed in rename operation. The join of X and Y (X⋈Y) can be represented as follows:(1)X⋈Y={x∪y|x∈X∧y∈Y∧Func(x∪y)}
where Func(x∪y) represents predicate and requires a minimum of one common attribute between *x* and *y*. It can be observed from the equation that natural joining combines both datasets, which introduces more redundant and irrelevant data. In order to avoid this situation, we used the left-join, which results in all the tuples of X and common attributes between X and Y. It is worth mentioning here that X contains the UNSW-NB15, which has more attacks as compared to NSL-KDD. One primary reason is to have more attacks after the fusion process. The left join of X and Y (X⟕Y) can be represented as follows:(2)$$X⟕Y=\left(X⋈Y\right)\cup \left(\left(X-\Pi_{x_{1},x_{2},\ldots x_{n}}\left(X⋈Y\right)\right)\times \left(\omega_{1},\omega_{2},\ldots \omega_{k}\right)\right)$$
where Π represents the projection of attributes ($x_{1},x_{2},\ldots x_{n}$) and ω shows the null values. The algorithm for the data fusion in *MIND* is represented in Algorithm 1 where the resultant fusion feature provides all the attributes of X as well as the common attributes between X and Y, as shown in Table 2 and Table 3.

In order to perform the data fusion procedure among the NSL-KDD and UNSW-NB15 datasets, we first create a database for data manipulation and generate two tables in that database to import NSL-KDD and UNSW-NB15 in their respective tables. These tables are then merged by using left join. The attributes which are common in these datasets are protocol and service. After the fusion of these datasets, a comprehensive dataset is produced which is further provided to the machine learning classifier. The overall procedure for data fusion scheme of the proposed approach is presented in Algorithm 1.
**Algorithm 1** Data Fusion1:**procedure** 
MIND2:    $X=\{x_{1},x_{2},x_{3},\ldots ,x_{n}\}$ attributes of Relation X3:    $Y=\{y_{1},y_{2},y_{3},\ldots ,y_{k}\}$ attributes of Relation Y4:    $T_{X}$: total tuples in relation X5:    $T_{Y}$: total tuples in relation Y6:    $t_{x}$: each tuple in relation X7:    $t_{y}$: each tuple in relation Y8:    $X_{\gamma }$: common attribute of relation X9:    $Y_{\gamma }$: common attribute of relation Y10:    $F_{R}$: final fused relation11:    **for** <each $t_{x}$ in $T_{X}$> **do**12:        Flag = False13:        **for** <each $t_{y}$ in $T_{Y}$> **do**14:           **if** <$t_{x}$.$X_{\gamma }$ = $t_{y}$.$Y_{\gamma }$> **then**15:               $F_{R}\leftarrow$$t_{x}$, $t_{y}$16:               Flag = True17:           **end if**18:        **end for**19:        **if** <Flag = False> **then**20:           $F_{R}\leftarrow$$t_{x}$21:        **end if**22:    **end for**23:**end procedure**

### 4.4. Machine Learning Implementation

Machine learning algorithms are heavily used in NIDS for intrusion detection optimization. Each of these algorithms has its own advantages and disadvantages in terms of computational complexity and efficiency. The ensemble approach is used by combining multiple classifiers to obtain the optimal result, which is considered the most efficient approach in machine learning [10,11]. Figure 2 shows the functional design of the method. In this work, model building for classification goes over the following steps. The data preprocessing is performed on the fused dataset by converting character values into numeric values as each machine learning algorithms work with numeric data only. There are two different approaches for this purpose which are as follows: label encoding and one-hot encoding. In label encoding, a unique integer is assigned to each label based on alphabetical order. This approach is simple and straightforward, however, one major issue with this approach is that algorithms may misinterpret these numeric values. This issue is addressed by the one-hot encoder (used in this paper), where a new column is added against each category and binary values are assigned to each column. This model can efficiently detect anomalies from complex networks. This ensemble-based model (bagging approach) having KNN as base classifiers can efficiently detect anomalies from complex networks. The individual prediction of each classifier is aggregated to obtain the final prediction and it also reduces the variance. KNN uses the similarity matric, which is based on the euclidean distance between the other points in the dataset and an unknown point that can be represented as follows:(3)$$d\left(a,b\right)=\sqrt{\sum_{i=1}^{j}{\left(a_{i}-b_{i}\right)}^{2}}$$
where *a* and *b* represent the attributes of observations. This model is based on 10-fold cross-validation as the time complexity of each KNN classifier is O(N log N), which is better than other classifiers.

## 5. Experimental Setup

In this section, we discuss the experimental setup for *MIND*. We used the Intel Core i5 2.3 GHz Central Processing Unit (CPU), 4 GB Random Access Memory (RAM), and Centos 7 Operating System (OS) system for conducting experiments.

Since RDBMS are not able to handle massive data coming from multiple sources, the Hadoop MapReduce framework [37] is used for the data fusion process of NSL-KDD and UNSW-NB15. On top of Hadoop MapReduce, the Hive tool [38] is used to merge two datasets. Hadoop is an open-source framework run as a distributed application on commodity hardware for processing a massive volume of data. Hundreds and thousands of computers work together in parallel for computing and storing gigantic data. Two primary components of Hadoop v2.x include the Hadoop distributed file system (HDFS) and yet another resource locator (YARN) for storing and processing extensive amounts of data across different commodity hardware. The earlier version of Hadoop v1.x did not contain YARN in its framework. Hadoop v2.x introduces YARN in its framework ecosystem.

The hive tool is used to provide data analysis on HDFS, which is an alternative to any programming language. A two-hundred-piece code written in any programming language can be achieved in ten to fifteen lines of code in Hive. Hive works in the same manner as RDBMS. It stores the data in a tabular fashion and uses HiveQL (similar to SQL in RDBMS) to query data through Hadoop. Data analysis is also performed by using HiveQL, which is stored on different tables of Hive. It provides an easy solution, particularly for database programmers [39]. There are two types of joins in Hive which are reduce side join and map side join. The reduce side join is also called repartitioned sort merge join and is widely used. However, it requires sort and shuffle phase which sometimes introduces delay and overhead over the networks. The map side join requires strong prerequisites and performs join before data reach to map. It minimizes the burden of merging and sorting in the reduced and shuffle phase of the MapReduce procedure. Moreover, it optimizes the performance as task completion takes less time. This is one of the major reasons that we prefer map side join in this work.

In order to validate our proposed approach, we perform cross validation. The fused dataset is divided into ten segments $S_{\alpha }$ where α = {1, 2, 3, ..., 10}. We select the segments of fused data that are not from $S_{\alpha }$ as *Training Data*
$\left(Data_{train}\right)$ and data from $S_{\alpha }$ for *testing data*
$\left(Data_{test}\right)$ for the $\alpha^{th}$ cross-validation. The segments of $\left(Data_{train}\right)$ for $\alpha^{th}$ cross-validation can be represented as the following.
(4)$$Data_{train}^{\alpha }=\underset{\beta \in [1,10]\wedge \beta \neq \alpha }{⋃}S_{\beta }$$

The classifier is further trained on this data and $\left(Data_{test}\right)$ is tested on this training data. The machine learning-based classification model is built on Python programming language using the Scikit-learn library. Jupiter notebook is used through Anaconda distribution for writing Python language code. Scikit-learn library is specially designed for solving machine learning-based problems. The task is to check whether the network packet data contains intrusion, so it is a binary classification issue. While building the classification model, data preprocessing is performed using a one-Hot-encoder scheme which converts the character into numeric values. In addition, an efficient machine learning algorithm must be selected for detecting anomalous data from the network data packet. Most machine learning-based classification algorithms such as Naive Bayes, Artificial Neural Network, K-Nearest Neighbor, Logistic Regression, Support Vector Machine, and Decision Tree have advantages and disadvantages. The ensemble technique is better and provides more efficient results than any single machine learning algorithm.

In this work, the model is created using the bagging technique of an ensemble approach. The based classifier used for the bagging technique is K-nearest neighbor. Different values of K neighbors are tested to obtain optimal results. After extensive experiments, k = 5 provides the best results for model optimization. In this work, the pipeline approach of the Sciket-learn library is used for model building. The philosophy of pipeline is to render a chain of steps together instead of as individual steps. In the pipeline, the preprocessing and classification model building is combined in one step. Moreover, the cross-validation is performed at the process level instead of at the model level in pipelining. This model is trained by using 10-fold cross-validations on the training dataset. The calculation performed against each fold and mean of 10-fold of False Positive Rate (FPR), Detection Rate (DR), Accuracy (ACC), True Positive Rate (TPR), and F-Measure parameters is shown as a result of the evaluation.

## 6. Evaluation and Results

This section describes the evaluation of *MIND* where, first of all, evaluation matrices are discussed, which is followed by proof of concept to evaluate the effectiveness of the approach. Finally, we present the comparative analysis of *MIND* with existing studies.

### 6.1. Evaluation Matrices

The NIDS model is evaluated by using a statistical matrix that checks whether network data is normal or abnormal. The evaluation matrices used in this work are the following: *accuracy (ACC)*, *detection rate (DR)*, *false positive rate (FPR)*, *true positive rate (TPR)*, and *F-Measure*. The network IDS should exhibit good TPR, F-Measure, DR, and ACC but with lower FPR. In this work, the confusion matrix is used for evaluation purposes. Table 4 shows the confusion matrix binary classification.

*ACC* is the proportion of total instances that are correctly classified and of true negative (*TN*) and true positive (*TP*) to the total size of the dataset and can be calculated as follows.
(5)$$ACC=\frac{TP+TN}{TP+TN+FP+FN}$$

*DR* shows the ratio of correct classification of network data and is also referred to as precision. It is the ratio of *TP* to all the instances considered as an intrusion and can be calculated as follows.
(6)$$DR=\frac{TP}{TP+FP}$$

*FPR*, sometimes referred to as false alarm rate, is the proportion of the total instance of normal data considered as false-positive (*FP*) to the total normal instances of data set, which can be calculated as follows.
(7)$$FPR=\frac{FP}{TN+FP}$$

*TPR* shows the proportion of correct classification for a given class division by total instances, which includes *TP* and false-negative (*FN*) of that class and is also called a recall. It can be calculated as follows.
(8)$$TPR=\frac{TP}{TP+FN}$$

*F-Measure* uses *DR* and *TPR* together to find the evaluation criterion. The value of *F-Measure* is calculated by the harmonic mean of gained *DR* and *TPR*, which can be calculated as follows.
(9)$$F\text{-}Measure=2\times \frac{DR\times TPR}{DR+TPR}$$

This section discusses the achieved results from the experiments with their analysis to show the soundness of *MIND*. While constructing the method, 10-fold cross-validation approach is used for the fused dataset in this work, which is better for check checking model accuracy. In order to ensure repeatability, the experimentation was performed various times.

Table 5 shows the findings of ACC, DR, FPR, TPR, and F-Measure with and without fusion. It can be observed that the ACC with fusion is above 99% in all the experiments. However, the average ACC is 92.47% with a maximum value of 92.88% without fusion. Similar to ACC, the average DR is 92.39% without fusion, whereas it is more than 99% with fusion. In this work, FPR is considered a false alarm rate with a maximum of 0.44% value with an average of 0.29% after fusion. In contrast, the minimum value of FPR without fusion is 15.91%, with an average value of 17.6%. The average TPR without fusion is 97.24%, whereas the same parameter shows 99.85% after fusion. Finally, the F-Measure shows 99.82% and 94.60% with and without fusion, respectively, which shows the effectiveness of *MIND*.

### 6.2. Proof of Concept

Figure 3 depicts the average outcomes of each parameter with and without fusion. As previously mentioned, the UNSW-NB15 is kept in X which results in all the tuples of X and common tuples of X and Y during left-join process. Therefore, we select UNSW-NB15 dataset for comparison to ensure the proof of concept for data fusion. The x-axis shows the parameters, whereas the y-axis shows the percentage value against each parameter. In addition, the red line represents the average values without fusion and the average values with fusion are presented in the blue line. The ACC, DR, TPR, and F-Measure in each experiment are more than 99% and the averages of ACC, DR, TPR, and F-Measure of all experiments with fusion are 99.80%, 99.80%, 99.85% and, 99.82%, respectively. The FPR in each experiment is less than 0.45% and the mean value of FPR is 0.29%, which indicates the less false alarm rate. In contrast, the same parameters show less performance without fusion, which clearly shows the effectiveness of *MIND*. It can be concluded, from the results, that the feature level fusion among two datasets and pipelining technique for model building can be a significant reason for better results of *MIND*.

Figure 4 shows a graphical representation of TPR (sensitivity) and FPR (1-specificity) for binary classification, which is presented as Receiver Operation Characteristics (ROC) curve. The curve closer to the top left corner indicates better performance. It can be observed clearly in Figure 4a where the average value of the area under the curve is (AUC = 0.99) against the ratio 1:1. However, the average value of the area under the curve is lower in Figure 4b, which shows better performance after fusion.

### 6.3. Comparative Analysis

In this section, we discuss the comparative analysis of the proposed approach *MIND* with the existing studies. Several parameters are considered for this analysis: the average ACC, DR, TPR, FPR, F1-Measure, data fusion, and overfitting issue. The comparative analysis of *MIND* with existing studies are represented in Figure 5. Notably, the graph of each parameter is presented separately and all the graphs are scaled up to show the effectiveness of *MIND*.

The accuracy of *MIND* is compared with the existing studies which are Jiang et al. [19], Zhao et al. [16], Gao et al. [21], Benmessahel et al. [22], Wu et al. [18], Devan et al. [17], Hajisalem et al. [24], and Gu et al. [20]. It is observed that our approach outperforms existing studies with minor or significant differences, as shown in Figure 5a. Moreover, there is a minor difference between the proposed approachs Gu et al. [20] and Hajisalem et al. [24], but it outperforms the rest of the studies clearly. The comparison with Gu et al. [20] and Hajisalem et al. [24] is magnified and presented in the subgraph of Figure 5a. The accuracy of these studies is slightly more than 99%, whereas it is exceeding 99.5% in *MIND*.

Similarly, the detection rate is shown in Figure 5b where it can be observed that the proposed approach is outperforming existing studies that include Zhao et al. [16], Jiang et al. [19], Gao et al. [21], Benmessahel et al. [22], Devan et al. [17], Wu et al. [18], and Zhang et al. [15]. A comparison of *MIND* with Zhang et al. [15] and Wu et al. [18] is enlarged in the subgraph of Figure 5b where it can be observed that the detection rate of existing studies does not exceed 98%, whereas the proposed method has a detection rate of more than 99.5%.

The TPR is represented in Figure 5c with current literature studies, which include Zhao et al. [16], Jiang et al. [19], Gao et al. [21], Benmessahel et al. [22], Devan et al. [17], Wu et al. [18], and Zhang et al. [15]. It can be observed that the values of TPR vary between 94% to 98%; however, the TPR of the proposed method iexceeds 99%. A clear difference can be observed, however, and the comparison of *MIND* with [15,18] is expanded for a clearer view.

The comparison of FPR is shown in Figure 5d and is only discussed by four studies which include Zhao et al. [16], Benmessahel et al. [22], Gu et al. [20], and Hajisalem et al. [24]. There is a clear difference between existing studies except Hajisalem et al. [24] and *MIND*. In order to provide a comparison with this study, we have shown a magnified sub-graph where it can be observed that the FPR of the proposed approach is less than 0.3%. However, the FPR is 0.4% in [24]. It is worth mentioning here that all the existing studies do not focus on all the parameters. Therefore, we present a comparison of the individual parameter of each study.

The comparison of F-Measure is presented in Figure 6, where the proposed approach is compared with several studies that include Zhao et al. [16], Jiang et al. [19], Gao et al. [21], Benmessahel et al. [22], Devan et al. [17], Wu et al. [18], and Zhang et al. [15]. It can be observed that the F-Measure of the proposed approach is more than 99% and outperforms existing literature. Conclusively, the ACC, DR, TPR, FPR, and F1-Measure parameters of the *MIND* outperform other studies by intelligently applying the model classifier on the fused dataset.

The proposed approach outperforms existing studies and one possible reason could be the data fusion scheme proposed in this work. Another reason could be the feature selection which enhances the overall performance of the system. Although the proposed approach is accurate, it still produces some false alarm rater. One possible reason for this could be the neglected features during feature selection.

## 7. Conclusions

With the growth of Internet users, new attacks are also introduced. These attacks are critical for the overall efficiency and security of the network. NIDS are used to avoid these attacks, however, a false alarm is a significant challenge due to the high volume and uncertain data. To this end, this paper proposes *MIND* that is a multi-source data fusion scheme for intrusion detection. A data fusion mechanism is used in this approach, which can be obtained by merging data from different sources to obtain more informative and certain data. The data fusion is performed using the left join method of relational algebra. This process is performed with the help of Hive, which runs on top of Hadoop Map-Reduce. This consistent and precise data is further implemented in machine learning algorithms. The KNN algorithm is used for this purpose with ensemble learning, increasing the generalizability to combine weak learners and turning them into strong learners. The extensive simulation shows that the data fusion scheme improves the overall efficiency of IDS with less false alarm rate. In the future, the resource utilization of our approach can be optimized.

## Figures and Tables

**Figure 1 sensors-21-04941-f001:**
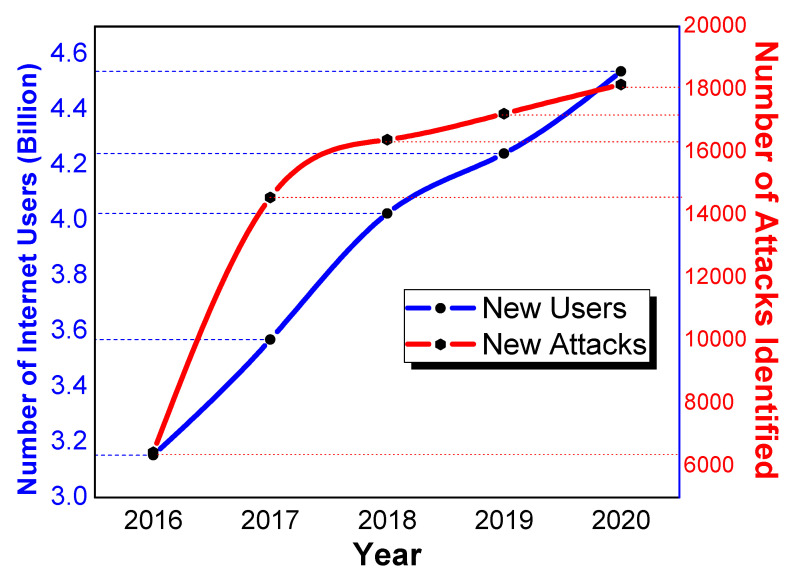
Number of Internet users and new attacks.

**Figure 2 sensors-21-04941-f002:**
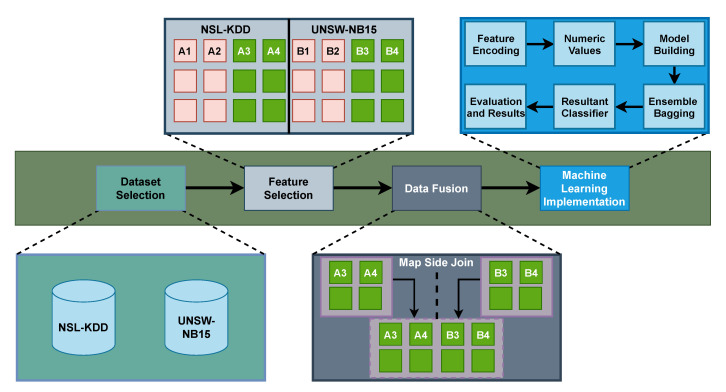
Working Design of Proposed Method.

**Figure 3 sensors-21-04941-f003:**
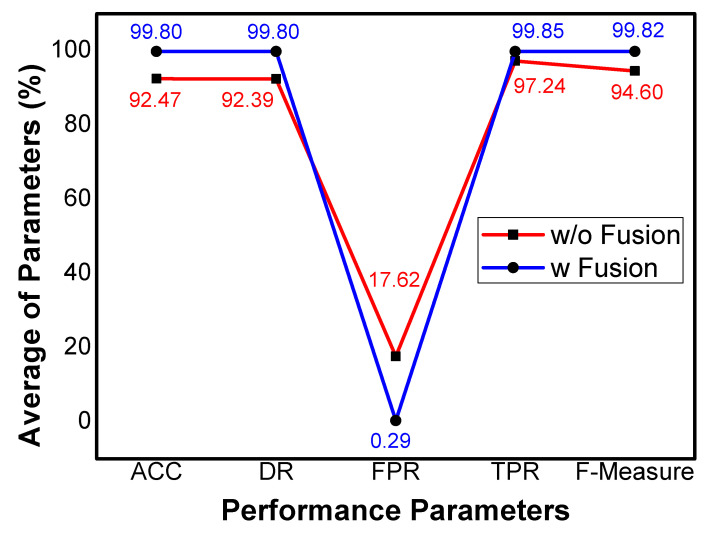
Average results of evaluation parameters.

**Figure 4 sensors-21-04941-f004:**
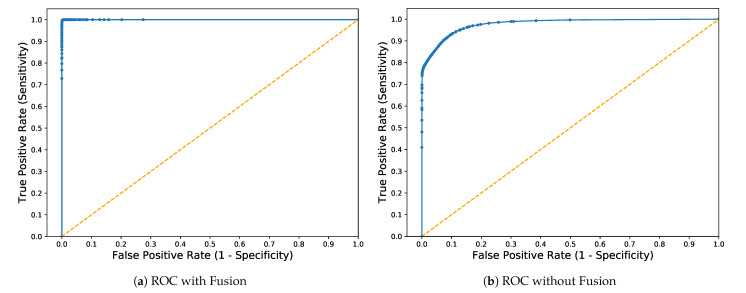
Receiver Operation Characteristics (ROC) Curve with and without Fusion.

**Figure 5 sensors-21-04941-f005:**
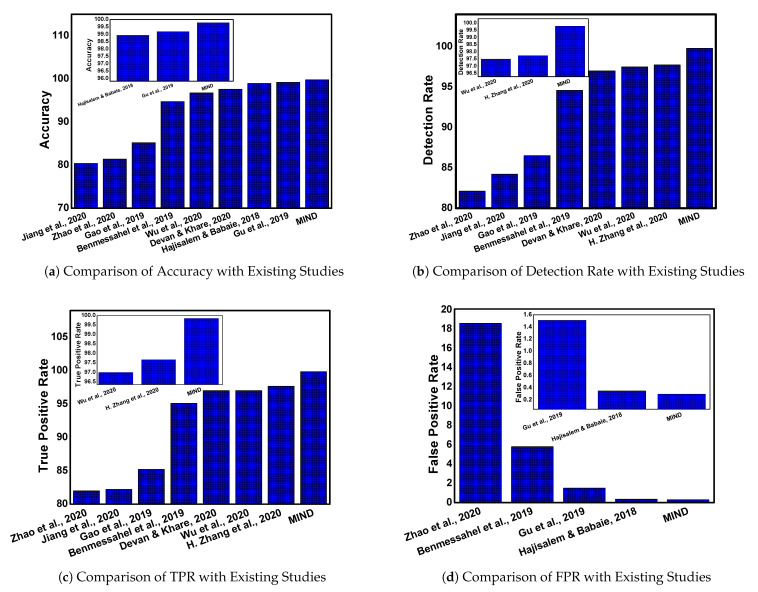
Comparative analysis with existing studies.

**Figure 6 sensors-21-04941-f006:**
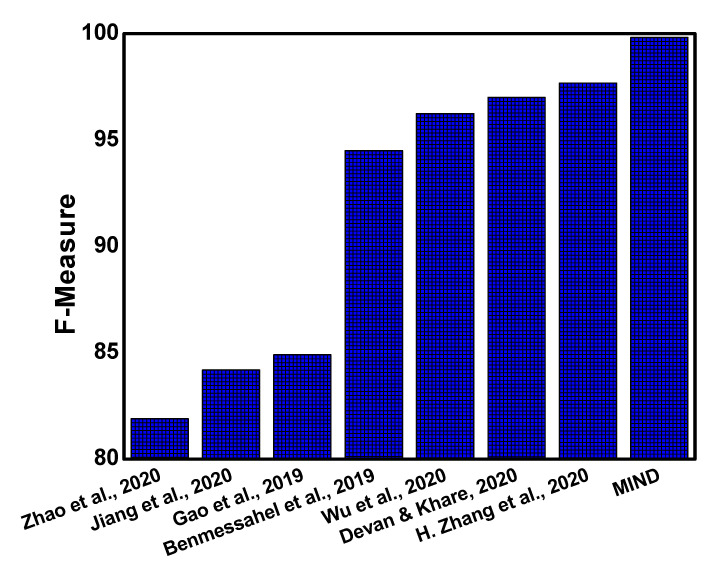
Comparison of F-Measure with existing studies.

**Table 1 sensors-21-04941-t001:** Comparison with existing studies.

Literature	Data Fusion	Over Fitting Issue	Evaluation Metrics
[15]	✗	✗	✗
[16]	✗	✗	✔
[17]	✗	✗	✗
[18]	✗	✔	✗
[19]	✗	✔	✗
[20]	✗	✔	✗
[21]	✗	✗	✗
[22]	✗	✔	✔
[23]	✗	✔	✗
[24]	✗	✔	✗
[25]	✔	✔	✗
[26]	✔	✗	✗
**MIND** (*This Work*)	✔	✔	✔

**Table 2 sensors-21-04941-t002:** Identified features for UNSW-NB15 dataset.

Index No	Feature Name
4	xServ
8	Sbytes
9	Dbytes
10	Rate
28	Smean
32	Ct_srv_src
36	Ct_dst_sport_ltm
42	Ct_srv_dst

**Table 3 sensors-21-04941-t003:** Identified features for NSL-KDD dataset.

Index No	Feature Name
3	Service
4	Source_bytes
5	Destination_bytes
6	Flag
23	Count
30	Srv_rerror_rate
33	Dst_host_srv_count
35	Dst_host_diff_srv_rate
37	Dst_host_srv_diff_host_rate
38	Dst_host_srv_serror_rate

**Table 4 sensors-21-04941-t004:** Confusion matrix for binary classification.

-	Normal	Abnormal
**Normal**	TN	FP
**Abnormal**	FN	TP

**Table 5 sensors-21-04941-t005:** Experimental results of evaluation parameters.

Experiments	ACC (%)		DR (%)		FPR (%)		TPR (%)		F-Measure (%)	
	w Fus.	w/o Fus.	w Fus.	w/o Fus.	w Fus.	w/o Fus.	w Fus.	w/o Fus.	w Fus.	w/o Fus.
1	99.88	92.86	99.85	92.76	0.22	16.16	99.79	97.10	99.82	94.88
2	99.83	92.88	99.85	92.54	0.22	16.73	99.86	97.38	99.86	94.90
3	99.78	92.58	99.77	92.35	0.33	17.14	99.96	97.14	99.82	94.69
4	99.85	91.57	99.86	92.85	0.20	15.91	99.89	96.97	99.88	94.87
5	99.86	92.61	99.88	91.01	0.17	20.52	99.88	97.25	99.76	94.01
6	99.71	92.81	99.69	92.43	0.44	16.94	99.83	97.10	99.76	94.71
7	99.72	92.56	99.71	94.71	0.42	16.86	99.82	97.34	99.81	94.85
8	99.78	92.49	99.78	92.28	0.32	17.34	99.84	97.20	99.76	94.67
9	99.72	92.46	99.70	92.13	0.43	17.70	99.83	97.63	99.87	94.63
10	99.84	92.86	99.85	92.83	0.21	20.92	99.88	97.27	99.88	93.84
**Average**	**99.80**	**92.47**	**99.80**	**92.39**	**0.29**	**17.62**	**99.85**	**97.24**	**99.82**	**94.60**

## Data Availability

Not applicable.

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
