# Peer review of "MIND: A Multi-Source Data Fusion Scheme for Intrusion Detection in Networks"

_sensors, 2021, doi:10.3390/s21144941_

Round 1
Reviewer 1 Report
• This paper is worth for acceptance, novelty of the idea seems interesting and small changes need to be incorporated in order to enhance.
• This paper deals with an exciting topic. The article has been read carefully, and some crucial issues have been highlighted in order to be considered by the author(s).
• All the acronyms should be defined and explained first before using them such that they become evident for the readers.
• Most of the typos and incorrect grammars have been corrected, but it is still necessary to subject the paper to proofreading.
• The paper needs to be restructured in order to be precise.The Introduction and related work parts give valuable information for the readers as well as researchers. In addition recent papers should be added in the part of related work.
• As it is real time application oriented, authors should care over the outcome of the proposed framework by meeting the future requirements too.
• Representation of figures needs to be improved.
• Grammatical errors should be validated.
• It would be good if similar domains [1], such as intrusion tolerant system, would be reflected in future research or related work.
[1] Kwon, Hyun, et al. "Optimal cluster expansion-based intrusion tolerant system to prevent denial of service attacks." Applied Sciences 7.11 (2017): 1186.
Author Response
We sincerely thank the reviewers for providing detailed comments, which have helped us in significantly improving the presentation and quality of our article. In the following, we provide a detailed response and changes made for each comment.
As the paper has gone through a revision, hence it is difficult to append verbatim changes for each response. We have categorically specified the places in the paper which have been changed at the end of each response.

Reviewer 2 Report
In the manuscript, the authors have proposed a multi-source data fusion-based network intrusion detection scheme, termed MIND. The overall organization of the manuscript is well. However, several issues need to be addressed before the acceptance.
- In terms of the related work, Table 1 illustrates the comparison between the existing works in the literature and the proposed MIND. However, the statement in the section is a little bit wordy and unorganized. Could the authors please improve the writing of Section 3 Related Work?
- In Line 194 of Page 5, the reference for NSL-KDD is a question mark [?].
- Please check the grammar of the sentence from Line 210 to Line 212 on Page 5.
- For Subsection 4.1, the dataset selection is not convinced, which seems like the authors subjectively select two datasets, namely, NSL-KDD and UNSW-NB15. Could the authors please provide more explanations? Why is the number of datasets used two? Why are these two?
- The sentence from Line 264 to Line 266 is quite confusing. Could the authors please rephrase the sentence in a more readable way?
- In terms of the experimental setup, the authors spare many words on the usage of Hadoop instead of the relational database. However, the info that the readers care about is ignored. How many items of data are used for training and testing, respectively?
- Furthermore, how do the authors set up the dataset for testing? There is no description of the testing dataset. However, the info of the testing dataset is significant, since it directly affects the testing results of the methods with and without data fusion. Without a proper testing dataset, the testing results are not persuasive enough.
- For the methods without data fusion, does it mean that the methods are trained with an individual dataset? Which dataset is used, NSL-KDD or UNSW-NB15?
- Equation (8) is incorrect.
- In terms of the evaluation and results, the numerical results are described without any insightful analysis.
- There are so many typos and misusage of words and punctuation marks, which harms the readability of the manuscript.
Author Response
Dear Reviewers,
We sincerely thank the reviewers for providing detailed comments, which have helped us in significantly improving the presentation and quality of our article. In the following, we provide a detailed response and changes made for each comment.
As the paper has gone through a revision, hence it is difficult to append verbatim changes for each response. We have categorically specified the places in the paper which have been changed at the end of each response.

Reviewer 3 Report
This paper presents a data-fusion approach to evaluate intrusion detection in networks. In particuar it has been presented how, merging two different types of data sources, performances using KNN increases.
The overall quality of the article is good, as the background study and the motivations are strongly described through all the Sections. In addition, the methods and the technology used are detailed as well. Results are interesting and seems to improve the reference values for the classification of those two datasets.
Unfortunately those are the only few things I can find interesting of this paper. As mentioned by the Authors in the "Related work" Section, there are other (almost ten) similar articles that uses Machine Learning to classify NSL-KDD and UNSW-NB15 datasets. Data fusion is not a novel approach, and here it is proposed (with some interesting results) just an application of that.
I think that authors should read the article twice to find some missing references on the text (such as in Section 4), and double check that those are uniform in the Bibliografy Section.
Author Response

(The authors gave the same response as above.)

Round 2
Reviewer 2 Report
Could the authors please double-check the usage of the semicolon? The mistakes are left unrevised.
Author Response
Dear Reviewer,
Thank you for your time and comments which help us to improve the overall quality of the manuscript.
All the unnecessary semicolons ( ; ) have been removed from the updated manuscript.